# On perturbation around closed exclusion processes

**Masataka Watanabe**

Graduate School of Informatics, Nagoya University, Nagoya 464-8601, Japan

max.washton@gmail.com

## Abstract

We derive the formula for the stationary states of particle-number conserving exclusion processes infinitesimally perturbed by inhomogeneous adsorption and desorption. The formula not only proves but also generalises the conjecture proposed in [Phys. Rev. E 97, 032135] to account for inhomogeneous adsorption and desorption. As an application of the formula, we draw part of the phase diagrams of the open asymmetric simple exclusion process with and without Langmuir kinetics, correctly reproducing known results.

## 1  Introduction

Among the famous solvable models of driven diffusive systems is the asymmetric simple exclusion process (ASEP). Aside from being solvable deep in the non-equilibrium regime, the model is interesting for its connections to various ideas in statistical physics such as boundary-induced phase transitions [1], the KPZ universality class [2], and random matrix theory [3]. It has also attracted attention for its wide applicability to various phenomena in physics, biology or society [4–6].

ASEP describes particles on a one-dimensional lattice that hop asymmetrically (*e.g.,* more frequently to the right than left). It is a continuous-time Markov process and as such it is interesting to study where the probability distribution settles after a long time, *i.e.,* its stationary distribution (state). This is equivalent to studying the eigenvector(s) of the Markov matrix with zero eigenvalue, which is possible for ASEP because of its $U_q(sl(2))$ symmetry [7–9]. However, remained elusive is the stationary distribution (as well as other properties) for exclusion processes without integrability, even though their practical usefulness and applicability are not impaired by the lack thereof.

One example of exclusion processes without integrability is the ASEP combined with Langmuir kinetics (ASEP-LK), where, in addition to asymmetric hopping, a particle can attach to or detach from the lattice at homogeneous rates. Even though the system has the $U_q(sl(2))$ symmetry on a periodic chain, it is indeed broken on a finite-length chain with boundary conditions. The model is proposed to describe unidirectional motion of motor proteins [10], so it is a good example of non-integrable exclusion processes with interesting applications.

One possible way to analytically study the properties of non-integrable exclusion processes is to take the thermodynamic limit. One can for example determine the phase diagram of the system by solving the fluid equations obtained by taking the coarse-grained, continuum limit. However, the strategy usually involves using the mean-field approximation, which may or may not be justified from first principles [10,11]. One can also derive the hydrodynamic equation by separating slow diffusion modes from fast transport modes as in [12]. Such analysis indeed gives the correct phase diagram at large volume, even though it does not account for fluctuations and so it does not constitute algorithmic computations of physical quantities. (However see [13] for the application of fluctuating hydrodynamics to an exclusion process.) It would therefore be desirable if we have another theoretical method with a clear regime of validity to compare with experiments or computer simulations. This could theoretically justify the mean-field approximations as well.

There is one such method which seemingly has been mostly overlooked – the perturbation theory. The ASEP-LK is a prime example: The stationary states of the periodic/closed/open ASEP have been obtained exactly, and so one can in principle obtain those of the ASEP-LK perturbatively when the ad/desorption rates are small. For example, [14] conjectured a formula for the stationary states of the closed ASEP with infinitesimally small Langmuir kinetics. This formula is yet to be proven despite having a simple and inviting form, however.

The goal of this paper is to set up such perturbation in generic situations. We consider perturbing a particle-number conserving hopping process with inhomogeneous ad/desorption and derive a formula for the stationary state at leading order. (The leading order result is meaningful because this is a degenerate perturbation theory.) Our formula potentially has various interesting applications. For one thing, the above-mentioned conjecture is its immediate consequence since closed ASEP is clearly a particle-number conserving process. We can also apply the formula to draw perturbative part of the phase diagrams of the open ASEP with/without Langmuir kinetics. We do so by interpreting the open boundary condition as a special case of the inhomogeneous ad/desorption acting only on boundary sites. The results, as we will see later, reproduce the results obtained in [11] but without relying on the mean-field approximation or any other unjustified approximations at all. Therefore we are going to have a clear regime of validity for our theoretical formula, even though the price we pay is the restriction to the perturbative regime.

The rest of the paper is organised as follows. We first briefly review Markov processes, in particular closed ASEP with/without infinitesimal Langmuir kinetics in Section 2. We then go on to construct the stationary states of generic particle-number conserving Markov processes infinitesimally perturbed by inhomogeneous ad/desorption in Section 3. This will, as a special case, prove the conjecture given in [14]. We will provide other applications of the formula by

deriving the phase diagram of the open ASEP with and without Langmuir kinetics in Section 4. We conclude in Section 5 with discussions and future directions.

## 2 Driven diffusive systems with ad/desorption

### 2.1 Continuous-time Markov process

Let us consider a continuous-time Markov process describing particles hopping on $L$ lattice sites (of arbitrary shapes and dimensions). We are interested in the time evolution of the probability associated with a given configuration. This is given by a collection of differential equations, conveniently written in matrix form using the Markov matrix $M$,

$$\frac{d}{dt}|P\rangle = M|P\rangle,$$

(2.1)

where $|P\rangle$ is a vector collecting probabilities of realising given configurations. In other words, by writing the configuration of particles as $(\tau_1, \ldots, \tau_L)$ where $\tau_i = 1$ ($\tau_i = 0$) means that a particle is (not) present at site $i$, and its realisation probability as $p(\tau_1, \ldots, \tau_L)$, we package the distribution into a state

$$|P\rangle = \sum_{\tau_1, \ldots, \tau_L} p(\tau_1, \ldots, \tau_L)|\tau_1, \ldots, \tau_L\rangle,$$

(2.2)

and this vector evolves according to the differential equation above.

For later convenience we denote the total Hilbert space as $V$, which is a tensor product of the Hilbert space $V_i$ on site $i$ from $i = 1$ to $L$,

$$V = \bigotimes_{i=1}^{L} V_i.$$

(2.3)

It can also be decomposed into a direct sum of fixed particle number subspaces $W_N$ (where $N$ indicates the number of particles in the system), so that

$$V = \bigoplus_{N=0}^{L} W_N.$$

(2.4)

Given such an evolution equation, one interest lies in finding where the probability distribution settles after a long time. This is given by the eigenvector of $M$ with eigenvalue zero. The number of such eigenvectors are expected to match that of the superselection sectors of $M$.

### 2.2 Closed exclusion process with ad/desorption

Our interest in this paper lies in the system $M$ such that $M = M_0 + \epsilon H$ where $\epsilon \ll 1$ is the perturbation parameter[1]. We require that $M_0$ conserves the particle number (*i.e.,* $U(1)$ symmetric) while $\epsilon H$ breaks it *via* ad/desorption. Concretely, we have

$$M \equiv M_0 + \epsilon H, \quad M_0\big|_{W_N} : W_N \to W_N$$

$$H = \sum_{i=1}^{L} h_i, \quad h_i \equiv \begin{pmatrix} -\alpha_i & \beta_i \\ \alpha_i & -\beta_i \end{pmatrix}_{V_i}.$$

(2.5)

---

[1]We will hereafter identify the Markov matrix as the corresponding system itself.

where $h_i$ only acts on $i$-sites, with $\epsilon \alpha_i$ and $\epsilon \beta_i$ representing adsorption and desorption rates at site $i$, respectively.

Note that, before perturbation, the state space breaks up into $L+1$ superselection sectors $W_N$, and we have a stationary state for each of them. We denote the $N$-particle stationary state as $|S_N\rangle$ hereafter. In addition, since the perturbation breaks the particle-number symmetry, there are no more superselection sectors for $M$. It therefore has only one stationary state, which we denote as $|\tilde{S}\rangle$.

The goal of this paper is to construct $|\tilde{S}\rangle$ in terms of $|S_N\rangle$ at leading order in $\epsilon$. It is by now clear that this is the zeroth order degenerate perturbation theory. We need to find the right basis on which the higher-order perturbation theory is run. We will study this in Section 3.

**Example: ASEP with Langmuir kinetics**   Before moving on, we present an example of such systems, known as the closed ASEP perturbed by Langmuir kinetics (ASEP-LK). The Markov matrix of closed ASEP-LK is given by the following,

$$M = M_0 + \epsilon H$$

$$M_0 = \sum_{i=1}^{L-1} M_{i,i+1}, \quad M_{i,i+1} = \begin{pmatrix} 0 & 0 & 0 & 0 \\ 0 & -q & 1 & 0 \\ 0 & q & -1 & 0 \\ 0 & 0 & 0 & 0 \end{pmatrix}_{V_i \otimes V_{i+1}}$$

$$H = \sum_{i=1}^{L} h_i, \quad h_i \equiv \begin{pmatrix} -\alpha & 1 \\ \alpha & -1 \end{pmatrix}_{V_i}.$$

(2.6)

where $M_0$ describes the closed ASEP and $\epsilon H$ the Langmuir kinetics. $M_{i,i+1}$ acts as an identity outside $V_i \otimes V_{i+1}$ and the bases of $M_{i,i+1}$ inside $V_i \otimes V_{i+1}$ are given by, from top to bottom columns or left to right rows, $|0,0\rangle$, $|0,1\rangle$, $|1,0\rangle$, and $|1,1\rangle$. The perturbation $\epsilon H$ describes a particle homogeneously detaching from the lattice at rate $\omega_d \equiv \epsilon$ while attaching at rate $\omega_a \equiv \epsilon \alpha$.

The closed ASEP, $M_0$, trivially conserves the particle number and so has superselection sectors labelled by it. There are therefore $L+1$ stationary states in $M_0$, which can be computed by using the Bethe ansatz as [15]

$$|S_N\rangle = \begin{bmatrix} L \\ N \end{bmatrix}_q^{-1} \sum_{(n)_N} q^{\sum_{j=1}^{N}(L-j+1-n_j)} |(n)_N\rangle,$$

(2.7)

where $|S_N\rangle$ denotes the $N$-particle stationary state. Here $\begin{bmatrix} L \\ N \end{bmatrix}_q$ is the $q$-binomial, defined by

$$\begin{bmatrix} L \\ N \end{bmatrix}_q \equiv \frac{(q;q)_L}{(q;q)_N (q;q)_{L-N}}, \quad (a;q)_n \equiv \prod_{i=1}^{n} (1 - aq^{i-1}),$$

(2.8)

and $(n)_N$ is an ordered collection of $N$ sites on which the particles are present. We also used a shorthand notation $|(n)_N\rangle$ to refer to the basis corresponding to such a configuration. The overall normalisation is because the sum of probabilities must be one.

Once we perturb the system by $\epsilon H$, the particle-number conservation is lost and there is only one stationary state, $|\tilde{S}\rangle$. Because the integrability is (mostly likely) lost due to the perturbation, it is considered difficult to derive the stationary state for this model. It was however conjectured in [14] that in the $\epsilon \equiv \omega_d \to 0$ limit (while fixing $\alpha$) $|\tilde{S}\rangle$ is given by

$$|\tilde{S}\rangle = \frac{1}{(1+\alpha)^L} \sum_{N=0}^{L} \binom{L}{N} \alpha^N |S_N\rangle + O(\epsilon).$$

(2.9)

We will prove this conjecture in the next section as a corollary to the main result.

# 3   Construction of the stationary state

We are going to prove the following theorem.

**Theorem 1.** *For a class of continuous-time Markov processes $M = M_0 + \epsilon H$ defined in (2.5), the stationary state of $M$ can be written in terms of the $N$-particle stationary states of $M_0$, $|S_N\rangle$, as*

$$|\tilde{S}\rangle \equiv \frac{1}{\sum_{N=0}^{L} p_N} \sum_{N=0}^{L} p_N |S_N\rangle + O(\epsilon), \quad p_N = \prod_{i=1}^{N}\left(\frac{A_L - A_{i-1}}{B_i}\right). \tag{3.1}$$

*where $A_N$ and $B_N$ are given by*

$$A_N \equiv \sum_{(n)_N} q[(n)_N] \sum_{n \in (n)_N} \alpha_n, \tag{3.2}$$

$$B_N \equiv \sum_{(n)_N} q[(n)_N] \sum_{n \in (n)_N} \beta_n, \tag{3.3}$$

*using*

$$|S_N\rangle \equiv \sum_{(n)_N} q[(n)_N] |(n)_N\rangle, \quad \sum_{(n)_N} q[(n)_N] = 1. \tag{3.4}$$

Incidentally, we have

$$A_0 = 0, \quad A_L = \sum_{i=1}^{L} \alpha_i, \quad B_0 = 0, \quad B_L = \sum_{i=1}^{L} \beta_i. \tag{3.5}$$

Before attempting to prove this theorem, we have one remark.

**Corollary 1.** *This, as a corollary, immediately proves the conjecture (2.9) given in [14].*

*Proof of Corollary 1.* Because $A_N = N\alpha$ and $B_N = N$ in the current case, we immediately have $p_N = \binom{L}{N}\alpha^N$. We therefore have $\sum_{N=0}^{L} p_N = (1 + \alpha)^L$. This concludes the proof of the conjecture (2.9). □

Now we move on to proving Theorem 1, but prior to this let us set up some notations which will be useful later. We denote $K_0$ as the subspace spanned by all the stationary states of $M_0$, while $K_1$ as the subspace spanned by all other eigenvectors. Because $M_0$ is non-normal, $K_0$ and $K_1$ are not orthogonal to each other.

Let us also present the strategy of the proof. We will be finding an eigenvector of a non-normal matrix in perturbation theory, starting from degenerate vacua. As the eigenvector we are looking for is the stationary state of the perturbed Markox matrix, we have

$$(M_0 + \epsilon H)|\tilde{S}\rangle = 0, \quad |\tilde{S}\rangle \equiv |S\rangle + \epsilon |v\rangle + O(\epsilon^2), \tag{3.6}$$

where $|S\rangle \in K_0$ and $|v\rangle \in K_1$. At order $O(\epsilon)$, the equation reduces to

$$H|S\rangle = M_0 |v\rangle \in K_1. \tag{3.7}$$

It might seem as if one needs to know all the eigenvectors of $M_0$ in order to impose such conditions, because the subspaces $K_{0,1}$ are not orthogonal to each other. However, this is too pessimistic. The space $K_1$ can be characterised by the fact that its inner product with the left

eigenvector of $M_0$ with vanishing eigenvalue is zero. In other words, if we write $|L_N\rangle$ as the $N$-particle eigenvector of $M_0^T$ (the transpose of $M_0$) with vanishing eigenvalue,

$$M_0^T |L_N\rangle = 0, \tag{3.8}$$

we have that

$$\langle L_N | \psi \rangle = 0 \quad \text{if} \quad |\psi\rangle \in K_1. \tag{3.9}$$

In addition, the form of $|L_N\rangle$ is immediate because $M_0$ is a Markov matrix,

$$|L_N\rangle = \sum_{(n_i)_N} |(n_i)_N\rangle. \tag{3.10}$$

This hinges on the fact that the sum of probabilities is constant in time and hence the sum of columns in a Markov matrix is zero (in each superselection sector, if any).[2] The normalisation is immaterial so we chose an arbitrary one.

Summarising the discussions above, we now need to find $|S\rangle \in K_0$ such that

$$\langle L_N | H | S \rangle = 0 \tag{3.11}$$

for any $N$. We parameterise $|S\rangle$ for convenience as

$$|S\rangle \equiv \frac{1}{\sum_{N=0}^L p_N} \sum_{N=0}^L p_N |S_N\rangle, \tag{3.12}$$

where we can set $p_0 = 1$. We also parameterise $|S_N\rangle$ as

$$|S_N\rangle \equiv \sum_{(n)_N} q[(n)_N] |(n)_N\rangle. \tag{3.13}$$

We demand that they are properly normalised, in other words that the sum of probabilities becomes one, $\sum_{(n)_N} q[(n)_N] = 1$.

Let us prove Theorem 1 now.

*Proof of Theorem 1.* First of all, $H|S_i\rangle$ does not overlap with $|L_N\rangle$ unless $i = N-1$, $N$, or $N+1$ because $H$ only takes $i$-particle states to $i$- or $(i \pm 1)$-particle states. Therefore the conditions $\langle L_N | H | S \rangle = 0$ reduce to a set of recursion relations,

$$p_{N-1} \langle L_N | H | S_{N-1}\rangle + p_N \langle L_N | H | S_N\rangle + p_{N+1} \langle L_N | H | S_{N+1}\rangle = 0, \tag{3.14}$$

where we set $p_{-1} = p_{L+1} = 0$ for consistency.

Let us now compute $\langle L_N | H | S_i\rangle$ for $i = N - 1$, $i = N$, and $i = N + 1$. Because we only need to compute the overlap with $|L_N\rangle$, we will only compute the projection of $H|S_i\rangle$ to $W_N$. Starting from $i = N - 1$, we have

$$H|S_{N-1}\rangle \Big|_{W_N} = \sum_{(n)_{N-1}} q[(n)_{N-1}] \sum_{n \notin (n)_{N-1}} \alpha_n |(n)_{N-1} \cup n\rangle, \tag{3.15}$$

---

[2]The form of $|L_N\rangle$ suggests that the overlap $\langle L_N | \psi \rangle$ is the sum of probabilities of realising $N$-particle states in $|\psi\rangle$. We thank Yuki Ishiguro and Jun Sato for discussions on this point. See also their paper [16] whose submission was coordinated with ours.

where $|(n)_{N-1} \cup n\rangle$ means adding a particle on site $n$ to the $(N-1)$-particle state $|(n)_{N-1}\rangle$. We then have

$$\langle L_N | H | S_{N-1}\rangle = \sum_{(n)_{N-1}} q[(n)_{N-1}] \sum_{n \notin (n)_{N-1}} \alpha_n \tag{3.16}$$

$$= \sum_{(n)_{N-1}} q[(n)_{N-1}] \left( \sum_{n=1}^{L} \alpha_n - \sum_{n \in (n)_{N-1}} \alpha_n \right) = A_L - A_{N-1}. \tag{3.17}$$

Let us continue to $i = N$. The $N$-particle subspace component in $H|S_N\rangle$ is given by

$$H|S_N\rangle \Big|_{W_N} = -\sum_{(n)_N} q[(n)_N] \sum_{n \notin (n)_N} \alpha_n |(n)_N\rangle - \sum_{(n)_N} q[(n)_N] \sum_{n \in (n)_N} \beta_n |(n)_N\rangle. \tag{3.18}$$

The overlap with $|L_N\rangle$ is hence given by

$$\langle L_N | H | S_N\rangle = -(A_L - A_N + B_N) \tag{3.19}$$

Finally we study the case where $i = N+1$. The $N$-particle subspace component in $H|S_{N+1}\rangle$ is given by

$$H|S_{N+1}\rangle \Big|_{W_N} = \sum_{(n)_{N+1}} q[(n)_{N+1}] \sum_{n \in (n)_{N+1}} \beta_n |(n)_{N+1} \setminus n\rangle \tag{3.20}$$

where $|(n_i) \setminus n\rangle$ means removing a particle on site $n$ from the $(N+1)$-particle state $|(n)_{N+1}\rangle$. The overlap with $|L_N\rangle$ is hence given by

$$\langle L_N | H | S_N\rangle = B_{N+1} \tag{3.21}$$

The recursion relation (3.14) therefore becomes

$$(A_L - A_{N-1})p_{N-1} - B_N p_N = (A_L - A_N)p_N - B_{N+1}p_{N+1}. \tag{3.22}$$

Since we have $(A_L - A_{N-1})p_{N-1} - B_N p_N|_{N=0} = 0$, we can derive a simplified recursion relation,

$$p_N = \frac{A_L - A_{N-1}}{B_N} p_{N-1}. \tag{3.23}$$

By solving this recursion relation, we conclude that the stationary state of the system $M$ becomes

$$|\tilde{S}\rangle \equiv \frac{1}{\sum_{N=0}^{L} p_N} \sum_{N=0}^{L} p_N |S_N\rangle + O(\epsilon), \quad p_N = \prod_{i=1}^{N} \left( \frac{A_L - A_{i-1}}{B_i} \right). \tag{3.24}$$

In other words we have successfully proven Theorem 1. □

## 4  Phase diagram of the open ASEP(-LK)

It is interesting to apply our formula to derive the phase diagram of the open ASEP with/without Langmuir kinetics in terms of perturbation theory. This can be done by considering the open

boundary condition as a particular case of the inhomogeneous ad/desorption. More conretely, the open ASEP-LK is defined by the following Markov matrix

$$M \equiv M_0 + \tilde{H}, \quad \tilde{H} = \sum_{i=1}^{L} \tilde{h}_i, \quad \tilde{h}_i \equiv \begin{pmatrix} -\omega_i^{[a]} & \omega_i^{[d]} \\ \omega_i^{[a]} & -\omega_i^{[d]} \end{pmatrix}_{V_i}, \tag{4.1}$$

where $M_0$ is the Markov matrix of the closed ASEP, while we demand $\omega_2^{[a]} = \omega_3^{[a]} = \cdots = \omega_{L-1}^{[a]}$ and $\omega_2^{[d]} = \omega_3^{[d]} = \cdots = \omega_{L-1}^{[d]}$. Note that the system becomes the open ASEP without Langmuir kinetics when $\omega_i^{[a]} = \omega_i^{[d]} = 0$ for $i = 2, \ldots, L-1$. When $\omega_i^{[a]}$ and $\omega_i^{[d]}$ are small, the system is amenable to perturbation theory and our formula (3.24) is applicable. We therefore set

$$\begin{aligned} \omega_1^{[a]} &= \epsilon \alpha, \quad \omega_L^{[d]} = \epsilon \beta \\ \omega_1^{[d]} &= \epsilon \gamma, \quad \omega_L^{[a]} = \epsilon \delta \\ \omega_i^{[a]} &= \epsilon a, \quad \omega_i^{[d]} = \epsilon b \quad \text{for } i = 2, \ldots, L-1 \end{aligned} \tag{4.2}$$

and compute the stationary state of the open ASEP-LK at leading order in $\epsilon \ll 1$. In other words, we have, in the language of (2.5),

$$\begin{aligned} \alpha_1 &= \alpha, \quad \alpha_2 = \cdots = \alpha_{L-1} = a, \quad \alpha_L = \delta \\ \beta_1 &= \gamma, \quad \beta_2 = \cdots = \beta_{L-1} = b, \quad \beta_L = \beta. \end{aligned} \tag{4.3}$$

For later convenience, we will denote the $N$-particle stationary state of the closed ASEP as

$$|S_N\rangle \equiv \sum_{(n)_N} q_L[(n)_N] |(n)_N\rangle, \quad q_L[(n)_N] = \begin{bmatrix} L \\ N \end{bmatrix}_q^{-1} q^{\sum_{j=1}^{N}(L-j+1-n_j)} \tag{4.4}$$

emphasising that the number of sites is $L$. We will also denote $q_L[(n)_N | \tau_1 = 0, 1, \tau_L = 0, 1]$ to restrict $(n)_N$ to obey particles at site $1$ and $L$ being present/absent. Equivalently, we can set $q_L[(n)_N | \tau_1 = 0, 1, \tau_L = 0, 1] = 0$ if $(n)_N$ is not consistent with $\tau_1 = 0, 1$ or $\tau_L = 0, 1$.

Let us now compute $A_i$ and $B_i$. We hereafter restrict our attention to $A_i$ only since $B_i$ can be obtained from $A_i$ by swapping $\alpha$ with $\gamma$, $\delta$ with $\beta$, and $a$ with $b$. We have

$$A_N = \sum_{\tau_1, \tau_L = 0, 1} A_N^{\tau_1, \tau_L}, \tag{4.5}$$

where (for example) $A_N^{0,1}$ means that the sum over $(n)_N$ in the definition of $A_N$ is restricted to its subset in which $\tau_1 = 0$ (absent) and $\tau_L = 1$ (present). More concretely, they are defined as

$$A_N^{\tau_1, \tau_L} \equiv A_N \equiv \sum_{(n)_N} q_L[(n)_N | \tau_1, \tau_L] \sum_{n \in (n)_N} \alpha_n. \tag{4.6}$$

This will not become too complicated as $q_L[(n)_N | \tau_1, \tau_L]$ can be related to $q_{L-2}[(n)_N]$, $q_{L-2}[(n)_{N-1}]$, etc. For example,

$$q_L[(n)_N | \tau_1 = 0, \tau_L = 0] \times \begin{bmatrix} L \\ N \end{bmatrix}_q = q^{2N-N} q_{L-2}[(n)_N] \times \begin{bmatrix} L-2 \\ N \end{bmatrix}_q, \tag{4.7}$$

where $2N$ and $-N$ in the exponent comes from the shifting of $L$ to $L-2$ and of $n_j$ to $n_j - 1$, respectively. Similar arguments lead to

$$q_L[(n)_N | \tau_1 = 0, \tau_L = 0] = \frac{\left[ \begin{smallmatrix} L-2 \\ N \end{smallmatrix} \right]_q}{\left[ \begin{smallmatrix} L \\ N \end{smallmatrix} \right]_q} q^N \times q_{L-2}[(n)_N],$$

$$q_L[(n)_N | \tau_1 = 0, \tau_L = 1] = \frac{\left[ \begin{smallmatrix} L-2 \\ N-1 \end{smallmatrix} \right]_q}{\left[ \begin{smallmatrix} L \\ N \end{smallmatrix} \right]_q} q^0 \times q_{L-2}[(n)_{N-1}],$$

$$q_L[(n)_N | \tau_1 = 1, \tau_L = 0] = \frac{\left[ \begin{smallmatrix} L-2 \\ N-1 \end{smallmatrix} \right]_q}{\left[ \begin{smallmatrix} L \\ N \end{smallmatrix} \right]_q} q^{L-1} \times q_{L-2}[(n)_{N-1}], \tag{4.8}$$

$$q_L[(n)_N | \tau_1 = 1, \tau_L = 1] = \frac{\left[ \begin{smallmatrix} L-2 \\ N-2 \end{smallmatrix} \right]_q}{\left[ \begin{smallmatrix} L \\ N \end{smallmatrix} \right]_q} q^{L-N} \times q_{L-2}[(n)_{N-2}].$$

By summing over $(n)_N$ in (4.6), we get, by noting that $\sum_{(n)_N} q_{L_0}[(n)_{N_0}] = \left[ \begin{smallmatrix} L_0 \\ N_0 \end{smallmatrix} \right]_q$,

$$A_N = A_N^{0,0} + A_N^{0,1} + A_N^{1,0} + A_N^{1,1}$$

$$A_N^{0,0} = \frac{\left[ \begin{smallmatrix} L-2 \\ N \end{smallmatrix} \right]_q}{\left[ \begin{smallmatrix} L \\ N \end{smallmatrix} \right]_q} q^N \times aN, \quad A_N^{0,1} = \frac{\left[ \begin{smallmatrix} L-2 \\ N-1 \end{smallmatrix} \right]_q}{\left[ \begin{smallmatrix} L \\ N \end{smallmatrix} \right]_q} q^0 \times (a(N-1) + \delta) \tag{4.9}$$

$$A_N^{1,0} = \frac{\left[ \begin{smallmatrix} L-2 \\ N-1 \end{smallmatrix} \right]_q}{\left[ \begin{smallmatrix} L \\ N \end{smallmatrix} \right]_q} q^{L-1} \times (\alpha + a(N-1)), \quad A_N^{1,1} = \frac{\left[ \begin{smallmatrix} L-2 \\ N-2 \end{smallmatrix} \right]_q}{\left[ \begin{smallmatrix} L \\ N \end{smallmatrix} \right]_q} q^{L-N} \times (\alpha + a(N-2) + \delta),$$

and likewise

$$B_N = B_N^{0,0} + B_N^{0,1} + B_N^{1,0} + B_N^{1,1}$$

$$B_N^{0,0} = \frac{\left[ \begin{smallmatrix} L-2 \\ N \end{smallmatrix} \right]_q}{\left[ \begin{smallmatrix} L \\ N \end{smallmatrix} \right]_q} q^N \times bN, \quad B_N^{0,1} = \frac{\left[ \begin{smallmatrix} L-2 \\ N-1 \end{smallmatrix} \right]_q}{\left[ \begin{smallmatrix} L \\ N \end{smallmatrix} \right]_q} q^0 \times (b(N-1) + \beta) \tag{4.10}$$

$$B_N^{1,0} = \frac{\left[ \begin{smallmatrix} L-2 \\ N-1 \end{smallmatrix} \right]_q}{\left[ \begin{smallmatrix} L \\ N \end{smallmatrix} \right]_q} q^{L-1} \times (\gamma + b(N-1)), \quad B_N^{1,1} = \frac{\left[ \begin{smallmatrix} L-2 \\ N-2 \end{smallmatrix} \right]_q}{\left[ \begin{smallmatrix} L \\ N \end{smallmatrix} \right]_q} q^{L-N} \times (\gamma + b(N-2) + \beta).$$

## 4.1  Phase diagram of the open ASEP

We are now ready to compute the stationary state of the open ASEP by setting $a = b = \gamma = \delta = 0$. From the above computations, we have

$$A_i = \alpha q^{L-N} \times \frac{1-q^N}{1-q^L}, \quad B_i = \beta \times \frac{1-q^N}{1-q^L}, \tag{4.11}$$

which leads to

$$p_N \equiv \prod_{i=1}^{N} \left( \frac{A_L - A_{i-1}}{B_i} \right) = \left( \frac{\alpha}{\beta} \right)^N \times \left[ \begin{smallmatrix} L \\ N \end{smallmatrix} \right]_q. \tag{4.12}$$

Therefore the stationary state $|\tilde{S}\rangle$ of the open ASEP becomes, at leading order in $O(\epsilon)$,

$$|\tilde{S}\rangle \equiv \frac{1}{\sum_{N=0}^{L}(\alpha/\beta)^N \times \begin{bmatrix} L \\ N \end{bmatrix}_q} \sum_{N=0}^{L} \left(\frac{\alpha}{\beta}\right)^N \times \begin{bmatrix} L \\ N \end{bmatrix}_q |S_N\rangle + O(\epsilon) \tag{4.13}$$

$$= \frac{1}{\sum_{N=0}^{L}(\alpha/\beta)^N \times \begin{bmatrix} L \\ N \end{bmatrix}_q} \sum_{N=0}^{L} \left(\frac{\alpha}{\beta}\right)^N \sum_{(n)_N} q^{\sum_{j=1}^{N}(L-j+1-n_j)} |(n)_N\rangle + O(\epsilon), \tag{4.14}$$

from which all relevant physical quantities (particle number density, $n$-point functions, etc.) can be computed. Incidentally, the normalisation constant can be written more compactly as

$$\sum_{N=0}^{L} \left(\frac{\alpha}{\beta}\right)^N \times \begin{bmatrix} L \\ N \end{bmatrix}_q = {}_2\phi_0 \begin{bmatrix} q^{-N}, 0 \\ \emptyset \end{bmatrix}; q, \frac{\alpha}{\beta} \times q^N \end{bmatrix}, \tag{4.15}$$

where ${}_r\phi_s$ is the $q$-hypergeometric function, defined as

$${}_r\phi_s \begin{bmatrix} a_1, a_2, \ldots, a_r \\ b_1, b_2, \ldots, b_s \end{bmatrix}; q, z \end{bmatrix} = \sum_{n=0}^{\infty} \frac{(a_1, a_2, \ldots, a_r; q)_n}{(b_1, b_2, \ldots, b_s, q; q)_n} \left((-1)^n q^{\binom{n}{2}}\right)^{s+1-r} z^n, \tag{4.16}$$

in which $(a_1, a_2, \ldots, a_r; q)_n \equiv \prod_{i=1}^{r}(a_i; q)_n$.

Let us now detect the phase transition in the open ASEP by computing the particle number density, or equivalently, the one point function $\langle \tau_i \rangle$. For the sake of simpler analytic computations, we hereafter restrict our attention to $q = 0$. This makes thing particularly easy because the particle number density $\langle \tau_i \rangle_N$ of $|S_N\rangle$ is given by the step function,

$$\langle \tau_i \rangle_N = \begin{cases} 1 & i \geq L - N + 1 \\ 0 & i \leq L - N \end{cases}. \tag{4.17}$$

The number density $\langle \tau_i \rangle$ of $|\tilde{S}\rangle$ is then given by (at leading order in $\epsilon$)

$$\langle \tau_i \rangle = \frac{\sum_{N=L-i+1}^{L}(\alpha/\beta)^N}{\sum_{N=0}^{L}(\alpha/\beta)^N} = \frac{(\alpha/\beta)^{L+1-i} - (\alpha/\beta)^{L+1}}{1 - (\alpha/\beta)^{L+1}}, \tag{4.18}$$

where we used $\lim_{q\to 0} \begin{bmatrix} L \\ N \end{bmatrix}_q = 1$. We plot $\langle \tau_i \rangle$ for some values of $\alpha/\beta$ in Figure 1.

Let us take the thermodynamic limit $L \to \infty$. It is immediate to see that the behaviour of $\langle \tau_i \rangle$ are completely different for three cases, $\alpha/\beta \lesseqgtr 1$. For $\alpha/\beta < 1$, we have

$$\langle \tau_i \rangle = \begin{cases} 0 & \text{for } L - i \gg L^0 \\ \left(\frac{\alpha}{\beta}\right)^{L+1-i} & \text{for } L - i = O(L^0) \end{cases}, \tag{4.19}$$

for $\alpha/\beta = 1$,

$$\langle \tau_i \rangle = \frac{i}{L+1}, \tag{4.20}$$

and for $\alpha/\beta > 1$,

$$\langle \tau_i \rangle = \begin{cases} 1 - \left(\frac{\alpha}{\beta}\right)^{-i} & \text{for } i = O(L^0) \\ 1 & \text{for } i \gg O(L^0) \end{cases}. \tag{4.21}$$

Corresponding to the number density in the bulk region of the open chain, we call the phase realised for $\alpha/\beta < 1$ as the low-density phase, $\alpha/\beta = 1$ as the coexistence phase, and $\alpha/\beta > 1$ as the high-density phase. This is consistent with the known results obtained using exact methods in [17]. We depict our perturbative phase diagram of the open ASEP in Figure 2.

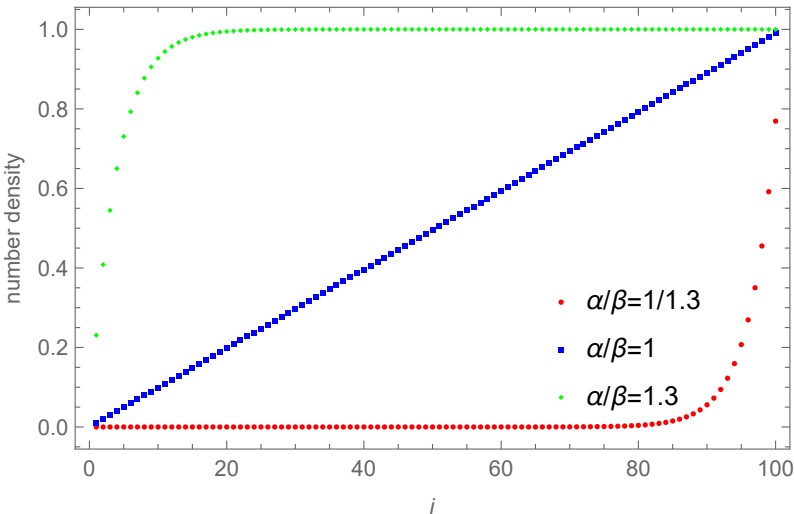

Figure 1: Plot of the particle number densities of the open ASEP (at $q = 0$) as functions of lattice sites $i$. We take the number of lattice sites to be $L = 100$. For $L$ as large as $100$, we already see three distinct phases – $\alpha/\beta < 1$ corresponds to the low-density phase, $\alpha/\beta = 1$, the coexistence phase, and $\alpha/\beta > 1$, the high-density phase.

## 4.2   Phase diagram of the open ASEP-LK

We can also compute the stationary state of the open ASEP-LK by turning on $a$ and $b$. Just as in the case of the open ASEP, we have

$$A_i = \frac{a(i-1) + aq^i + (\alpha - a)q^{L-i} + (a - \alpha - ai)q^L}{1 - q^L},$$  (4.22)

$$B_i = \frac{b(i-1) + \beta + (b - \beta)q^i - bq^{L-i} + (b - bi)q^L}{1 - q^L},$$  (4.23)

from which we can compute the stationary state of the open ASEP-LK at leading order in $\epsilon$. In particular at $q = 0$, $p_N$ can be expressed concisely as

$$p_N = \left(-\frac{a}{b}\right)^n \frac{\left(-L - \frac{\alpha}{a} + 1\right)_n}{\left(\frac{\beta}{b}\right)_n},$$  (4.24)

where $(x)_n \equiv \prod_{i=0}^{n-1}(x + i)$ is the Pochhammer symbol. One can then compute $\langle \tau \rangle_i = \sum_{L+1-i}^{L} p_N / \sum_{0}^{L} p_N$ and express it using hypergeometric functions, but we will not discuss this further as it will just be unnecessarily complex. We plot $\langle \tau \rangle_i$ for some parameters in Figure 3.

We now take the thermodynamic limit, $L \to \infty$. For the sake of manageability we will only consider the bulk region of the open chain, so that we take $i \to \infty$ at the same time while fixing $x \equiv i/L$. We will also take $a, b \to 0$ while fixing $\Omega_a \equiv aL$ and $\Omega_b \equiv bL$ – otherwise the collective effect of the bulk ad/desorption will dominate the physics and there will be no interesting phase transitions.

Let us compute $\langle \tau \rangle_i = \sum_{N=L+1-i}^{L} p_N / \sum_{N=0}^{L} p_N$. At large $L$ and at fixed $x$, $\Omega_a$, $\Omega_b$, it simply becomes

$$\rho(x) \equiv \langle \tau \rangle_i = \begin{cases} 1 & \text{when } p_{L-i+1}/p_{L-i} > 1 \\ 0 & \text{when } p_{L-i+1}/p_{L-i} < 1 \end{cases},$$  (4.25)

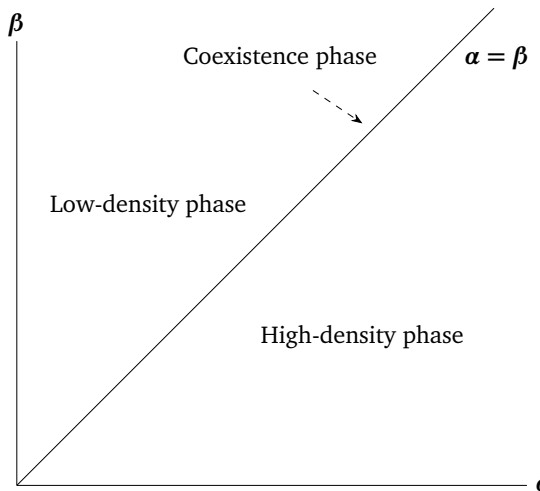

Figure 2: The phase diagram of the open ASEP. The horizontal axis represents the adsorption rate at site $i = 1$, while the vertical, the desorption rate at site $i = L$. This recovers the perturbative part of the known phase diagram of the open ASEP, obtained exactly in [17].

where we have

$$\frac{p_{L-i+1}}{p_{L-i}} = \frac{\Omega_a x + \alpha}{\Omega_b (1-x) + \beta} + O(L^{-1}), \tag{4.26}$$

for general $0 < q < 1$. This means that the domain-wall that separates the low- and the high-density phase happens at $x_d$ (the former appears for $x < x_d$ and the latter, $x > x_d$), given by

$$x_d = \frac{\Omega_b - \alpha + \beta}{\Omega_a + \Omega_b}. \tag{4.27}$$

We call this the domain-wall phase (called the shock phase in [11]).[3]  Additionally, when $x_d > 1$, the system is in the low-density phase, whereas when $x_d < 0$, it is in the high-density phase.  Summarising this, we have the low-density phase when $\beta > \alpha + \Omega_a$, the domain-wall phase when $\alpha - \Omega_b < \beta < \alpha + \Omega_a$, and the high-density phase when $\beta < \alpha - \Omega_b$. This is consistent with the results obtained using the (theoretically unjustified but numerically confirmed) mean-field approximation in [11]. We depict our perturbative phase diagram of the open ASEP-LK in Figure 4.

## 5   Discussions and outlook

In this paper, we studied the effect of perturbation on generic closed exclusion processes. We first derived the formula that expresses the stationary state of closed processes (infinitesimally) perturbed by ad/desorption in terms of that of the unperturbed system. The rates of ad/desorption did not have to be homogeneous in sites, so as a consequence we proved the formula in [14] while generalising it. We pointed out that our formula is a result of the simple degenerate perturbation theory on non-normal matrices.

---

[3]The position of the domain wall $x_d$ is indeed consistent with numerics, see Figure 3. We expect the position to lie at $i = 70, 50, 30$ for $\alpha = 13, 15, 17$, respectively.

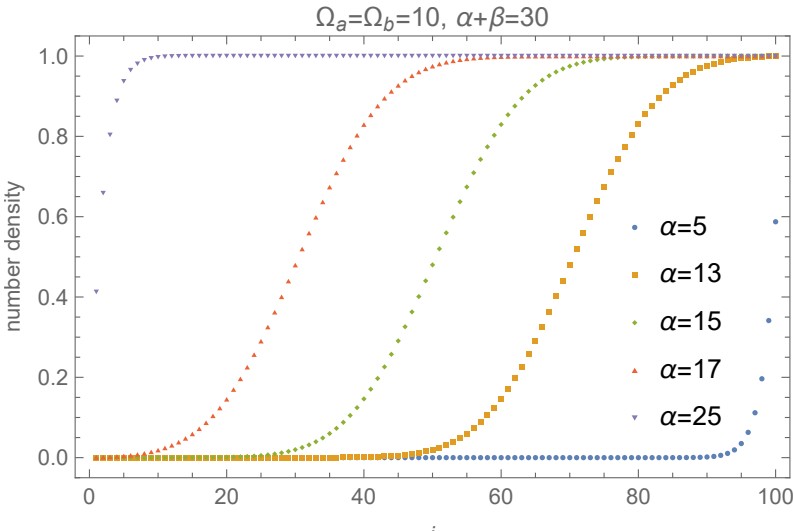

Figure 3: Plot of the particle number densities of open ASEP-LK (at $q = 0$) as functions of lattice sites $i$. We take the number of lattice sites to be $L = 100$. We set $\Omega_a \equiv aL = 10$, $\Omega_b \equiv bL = 10$ and varied $\alpha, \beta$ while fixing $\alpha + \beta = 30$. We sampled $\alpha = 5, 13, 15, 17, 25$ in the plot. We see that $\alpha = 5$ is in the low-density phase, $\alpha = 13, 15, 17$, the domain-wall phase, and $\alpha = 25$, the high-density phase, consistent with analytic computations.

As an application of the formula, we drew the perturbative part of the phase diagram of the open ASEP(-LK), which agreed with known results. For the open ASEP we recognised three distinct phases, called the low-density, the coexistence, and the high-density phases. For the open ASEP-LK, on the other hand, we recognised the low-density and the high-density phases, as well as the domain-wall phase in which the system contains a domain wall separating the low- and the high-density regions. It is important that these results were obtained without using any theoretically unjustified approximations – we exactly know when and how much our approximation breaks down.

There are a number of interesting future directions. First of all, it would be interesting to continue the perturbation theory to higher orders in $\epsilon$. For example, if we compare the phase diagram of [11] with ours, we notice that the phase boundaries are not exactly straight, *i.e.,* $\beta$ at the critical value is not a linear function of $\alpha$. It would be beneficial to compute the form of the phase boundaries at higher orders in perturbation theory to explain this.

Secondly, it would be interesting to apply our method to other systems of interest. For example, it would be interesting to apply it to the multi-lane ASEP [18] or to the ASEP(-LK) with inhomogeneous hopping rates [19].[4] It would also be interesting to study the open ASEP-LK by starting the perturbation from the exactly known stationary state of the open ASEP. Note that what we would need to do is in general non-degenerate perturbation theory. The result would allow us draw wider region of the phase diagram upon taking the thermodynamic limit. In particular, observing the three-phase coexistence predicted in [11] would be very interesting.

Studying the relaxation dynamics in perturbation theory is also interesting. One could, for example, compute the low-lying spectra and the corresponding states for the same class of theories at leading order in perturbation theory. In fact, [14] conjectures such a formula for

---

[4]Studying the latter in relation to sine-square deformation and other similar deformations [20, 21] might be also interesting.

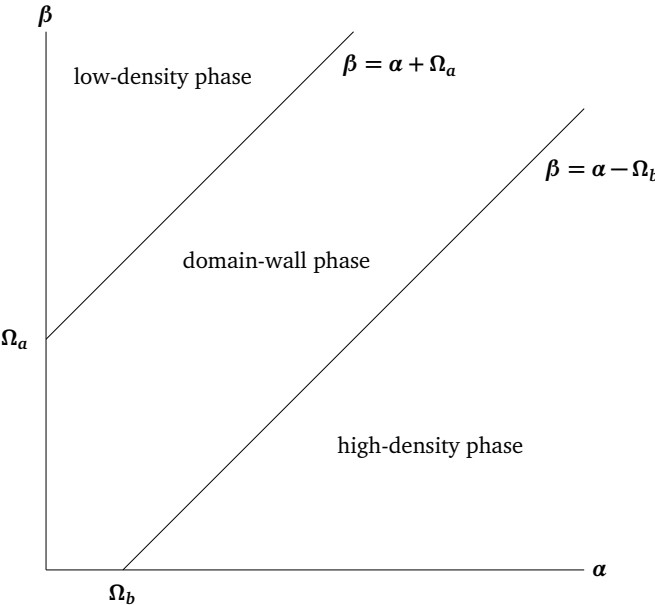

Figure 4: The phase diagram of the open ASEP-LK. The horizontal axis represents the adsorption rate at site $i = 1$, while the vertical, the desorption rate at site $i = L$. This recovers the perturbative part of the known phase diagram of the open ASEP-LK, obtained using the mean-field approximation in [11].

the closed ASEP-LK, so it would be interesting to start by proving it.

It would also be important to justify the hydrodynamic description theoretically. This could be either justifying the mean-field approximation or continuing the idea developed in [12]. In terms of the former, one could for example compute the two-point functions perturbatively in $\epsilon$; If they factorise in the thermodynamic limit, the mean-field approximation is justified at least perturbatively. In this context, it might be worthwhile to rewrite the open ASEP-LK in the language of one-dimensional (non-Hermitian) spin chains. The mean-field approximation can then be justified when the model flows to the free fixed-point in the infrared. In terms of the latter, it would be interesting to come up with a model which is strongly-coupled in the infrared, where the mean-field approximation cannot be justified but the hydrodynamic description is available [22–25]. Incidentally, in terms of the field theoretic understanding of the exclusion processes, interpreting the asymmetric hopping parameter $q$ as an imaginary vector potential is also interesting [26]. Because the $q \to 0$ limit corresponds to the limit of large imaginary vector potential, one might be able to use effective field theory to study such regions [27–33].

Lastly, studying the relationship between the general solvable exclusion processes with other models with $U_q(sl(2))$ symmetry would be interesting. In particular, the SYK model (a quantum mechanical model with all-to-all random interactions of $N$ fermions) in the double-scaling limit [34–36] is known to possess such a symmetry and it would be interesting to connect them further. It would also be interesting to interpret it in terms of Jackiw-Teitelboim gravity [37, 38], which is believe to be dual to the SYK model in the context of the AdS/CFT correspondence [39].

## Acknowledgements

The author is grateful to Yuki Ishiguro, Jun Sato, and Kei Tokita for fruitful discussions. The author is supported by Grant-in-Aid for JSPS Fellows No. 22KJ1777 and by MEXT KAKENHI Grant No. 24H00957.

## A   A "physical" proof of Theorem 1

We give an equivalent but more physical proof of Theorem 1. Note that the idea of this proof originally appears in [16], whose submission was coordinated with ours.

The idea of the proof is to interpret (3.11) as defining the stationary state of a new Markov process by using the fact that $\langle L_N|\psi\rangle$ is the sum of probabilities of realising $N$-particle states in $|\psi\rangle$. Indeed, $\langle L_N|H|S\rangle = 0$ means that the probability distribution labelled by the number of particles (forgetting the details about where individual particles are) in the system is unaltered by the action of $H$. We will denote the state representing the sum of all $N$-particle states as $\widehat{N}$ and its realisation probability as $\tilde{p}_N$. Now, the action of $H$ is such that it takes $\widehat{N}$ to $\widehat{N+1}$ with probability $A_L - A_N$ per unit time and $\widehat{N}$ to $\widehat{N-1}$ with $B_N$. Then, for $\tilde{p}_0$ to be constant in time, we have $B_1\tilde{p}_1 = (A_L - A_0)\tilde{p}_0$, for $\tilde{p}_1$, $B_2\tilde{p}_2 = (A_L - A_1)\tilde{p}_1$, and so on, up to $B_L\tilde{p}_L = (A_L - A_{L-1})\tilde{p}_{L-1}$. Equivalently, we have

$$\tilde{p}_N = \frac{A_L - A_{N-1}}{B_N}\tilde{p}_{N-1}. \tag{A.1}$$

We already know from linear algebra that $|S\rangle$ needs to be spanned only by using $|S_N\rangle$, as in (3.12). Therefore $\tilde{p}_N$ must be identified with $p_N / \sum_{N=1}^{L} p_N$. This concludes another proof of Theorem 1.

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
