# Peer review of "On perturbation around closed exclusion processes"

_SciPost Physics_

## Round 2 · Referee Report · Anonymous (Referee 1) · 2024-8-24

Report

In the revised manuscript, the requested changes 1 and 3 of my first report have been properly done by the Author.
Concerning the requested change 2, I think that there is still an inconsistency in the manuscript.
The source of this is that two conflicting normalizations are used for q[(n)_N]. Using the definition in Eq. (4.4), the sum over (n)_N gives 1. See also the sentence below Eq. (3.13).
Now in the revised manuscript, a new sentence is added above Eq. (4.9), which says that the sum is [L N]_q instead of 1. But it must be 1, if we stick to the definition of q[(n)_N] in Eq. (4.4).
If we use the definition in Eq. (4.4) (which implies a normalization to 1), then Eqs. (4.7) and (4.8) are wrong, as the ratio of binomial coefficients are missing here (the same ratios as appear in Eqs. (4.9)).
If this point is corrected, the manuscript will be suitable for publication.

Recommendation

Ask for minor revision

---

## Round 2 · List of Changes

As a reply to the first referee:

Requested change 1: Thank you for encoiuraging me to do it. Rereading my manuscript, indeed this part was not very clearly written. I changed the draft so that it is self-contained. I added a paragraph on page 5, starting from "Let us also present the strategy of the proof."

Requested change 2: This was a bad explanation of mine. We are meant to sum over (4.8) on the LHS of (4.6), so we use the fact that $\sum_{(n)_N}q_{L_0}[(n)_{N_0}]=\binom{L_0}{N_0}_q$ . Then we will get the q-binomials in the final expressions. I changed the draft accordingly, around (4.9).

Requested change 3: This is very important. Thank you! I changed some parts of the draft where I over-emphasized the mean-field approximations. I also cited related papers I found, as well as Phys. Rev. E 67, 066117 (2003).

I added the following on page 3,

"One can also derive the hydrodynamic equation by separating slow diffusion modes from fast transport modes as in \cite{cond-mat/0302208}.
Such analysis indeed gives the correct phase diagram at large volume, even though it does not account for fluctuations and so it does not constitute algorithmic computations of physical quantities. (However see \cite{1803.06829} for the application of fluctuating hydrodynamics to an exclusion process.)"

and on page 14,

"It would also be important to justify the hydrodynamic description theoretically.
This could be either justifying the mean-field approximation or continuing the idea developed in \cite{cond-mat/0302208}.
In terms of the former, one could for example compute the two-point functions perturbatively in ϵ;
If they factorise in the thermodynamic limit, the mean-field approximation is justified at least perturbatively.
In this context, it might be worthwhile to rewrite the open ASEP-LK in the language of one-dimensional (non-Hermitian) spin chains.
The mean-field approximation can then be justified when the model flows to the free fixed-point in the infrared.
In terms of the latter, it would be interesting to come up with a model which is strongly-coupled in the infrared, where the mean-field approximation cannot be justified but the hydrodynamic description is available \cite{1801.08952,2104.14650,2208.02124,2405.19984}."

As a reply to the second referee:

(a) Thank you! Everything should be fixed now.

(b) Indeed such a limit can be taken! For example, if we restrict to q=0 which is the simplest, one can set $\Omega_{a,b}=0$ in (4.26) to find $p_{L-i+1}/p_{L-i}=\alpha/\beta$. I however decided not to include this in the draft as I thought it would make the draft more complicated.

(c) This is very interesting. Let me come back to this in future followups.

(d) Additionally, I addressed the weakness by adding Appendix A, which is the physical version of the proof in the main text.

---

## Round 3 · List of Changes

I finally understood the referee's comment and fixed (4.7). Thank you so much!

---

## Editorial Decision

resubmitted